# Tardigrade small heat shock proteins can limit desiccation-induced protein aggregation

Jonathan D. Hibshman [1][✉], Serena Carra [2] & Bob Goldstein[1,3]

Small heat shock proteins (sHSPs) are chaperones with well-characterized roles in heat stress, but potential roles for sHSPs in desiccation tolerance have not been as thoroughly explored. We identified nine sHSPs from the tardigrade *Hypsibius exemplaris*, each containing a conserved alpha-crystallin domain flanked by disordered regions. Many of these sHSPs are highly expressed. Multiple tardigrade and human sHSPs could improve desiccation tolerance of *E. coli*, suggesting that the capacity to contribute to desicco-protection is a conserved property of some sHSPs. Purification and subsequent analysis of two tardigrade sHSPs, HSP21 and HSP24.6, revealed that these proteins can oligomerize in vitro. These proteins limited heat-induced aggregation of the model enzyme citrate synthase. Heterologous expression of HSP24.6 improved bacterial heat shock survival, and the protein significantly reduced heat-induced aggregation of soluble bacterial protein. Thus, HSP24.6 likely chaperones against protein aggregation to promote heat tolerance. Furthermore, HSP21 and HSP24.6 limited desiccation-induced aggregation and loss of function of citrate synthase. This suggests a mechanism by which tardigrade sHSPs promote desiccation tolerance, by limiting desiccation-induced protein aggregation, thereby maintaining proteostasis and supporting survival. These results suggest that sHSPs provide a mechanism of general stress resistance that can also be deployed to support survival during anhydrobiosis.

[1] Biology Department, University of North Carolina at Chapel Hill, Chapel Hill, NC, USA. [2] Department of Biomedical, Metabolic, and Neural Sciences, University of Modena and Reggio Emilia, Modena, Italy. [3] Lineberger Comprehensive Cancer Center, University of North Carolina at Chapel Hill, Chapel Hill, NC, USA. [✉]email: hibsjo01@email.unc.edu

Desiccation is an extensive stress on cells and organisms[1,2]. One of the subcellular consequences of drying is protein denaturation and aggregation, which can lead to loss of enzymatic function[3–5]. Molecular shield proteins have been proposed to bind partially unfolded proteins during desiccation to limit contact with other denatured proteins and the formation of toxic aggregates[3,6]. Late embryogenesis abundant (LEA) proteins from desiccation-tolerant organisms are one family of protectants that have been shown to fulfill this role[4,7–11]. A limited number of other proteins have also been shown to act as molecular shields, but there are likely many other proteins that carry out a similar function to prevent desiccation-induced aggregation and loss of protein function[6,12,13].

Small heat shock proteins (sHSPs) are small (12–43 kDa) proteins that contain an alpha-crystallin domain flanked by disordered N- and C-terminal sequences[14,15]. The regions flanking the alpha-crystallin domain often contribute to oligomerization of the protein—a feature that is conserved amongst many sHSPs and can be important for their function[16–19]. sHSPs act as chaperones to limit protein aggregation at high temperatures, and sHSP mutations have been associated with several human diseases[20–25]. Accordingly, many of these proteins are upregulated during stress[26,27]. In contrast to other heat shock proteins like HSP70, sHSPs are not ATP-dependent, instead functioning as "holdases" to limit protein aggregation until unfolded clients can either be refolded or degraded—a function akin to that of proposed molecular shield proteins during desiccation[28]. Plant sHSPs have been shown to improve osmotic stress survival and contribute to drought tolerance[29–32]. However, a role for sHSPs in animal desiccation tolerance has been less well-explored.

sHSPs have been reported to be highly abundant and upregulated during desiccation in multiple anhydrobiotic animals[33–37]. A small heat shock protein, p26, from the brine shrimp *Artemia* is very highly expressed in embryonic cysts that are destined for diapause[38]. p26 constitutes approximately 15% of the total non-yolk protein in these cysts, which are resistant to multiple stresses including anoxia, heat, and desiccation[33]. p26 improves thermotolerance of brine shrimp and heterologous expression can increase heat shock survival of bacteria[39]. In vitro, p26 could protect the enzyme citrate synthase from thermal inactivation[40]. Further, when combined with trehalose, p26 could improve viability of dried human cells[41]. In the tardigrade *Milnesium tardigradum*, a sHSP was highly abundant in both active and anhydrobiotic animals[42]. In a study of the transcriptional response to desiccation in *C. elegans*, the most upregulated gene during desiccation, with over 700-fold induction, encodes for a small heat shock protein[37]. Additionally, two *C. elegans* sHSP mutants had reduced survival when desiccated[37]. To our knowledge, sHSPs have not been tested for a role in desiccation tolerance in vitro in the absence of trehalose, nor using tardigrade-derived sHSPs. We hypothesized that sHSPs may function to chaperone against desiccation-induced protein aggregation, acting as molecular shields to support desiccation tolerance.

Tardigrades are able to withstand extreme stresses including desiccation, yet the molecular mechanisms they employ to survive are only beginning to be understood[2,43–50]. We identified a family of sHSPs from the tardigrade *Hypsibius exemplaris* to specifically test the hypothesis that sHSPs can function as molecular shields and limit desiccation-induced protein aggregation. We analyzed published RNA-seq studies and saw that tardigrade sHSPs were highly expressed and in some cases upregulated by desiccation[48,49]. We found that multiple sHSPs, when expressed heterologously in bacteria, were sufficient to improve desiccation survival. We purified HSP21 and HSP24.6 and found that they formed large, polydisperse oligomeric complexes, similar to what has been reported for other members of the sHSP family in plants and mammals[14]. We demonstrated that HSP21 and HSP24.6 could function as chaperones to limit heat-induced aggregation of the model enzyme citrate synthase (CS) and aggregation of total soluble bacterial protein lysate. Further, these sHSPs limited desiccation-induced aggregation and loss of function of CS, suggesting a role as a molecular shield in protecting proteins during desiccation.

## Results

**A family of sHSPs in H. exemplaris**. We identified nine small heat shock proteins from the most recent genome annotation (v3.1.5) of the tardigrade *H. exemplaris* (Fig. 1a)[49]. We named each of the proteins by its predicted molecular weight (Supplementary Table 1). The protein sequences of the nine sHSPs we identified are consistent with conserved features of other sHSPs, i.e., with an alpha-crystallin domain (ACD) flanked by an N-terminal region and short C-terminal region[15]. Interestingly, three of the sHSP genes, HSP17, HSP19, and HSP20, are located next to each other on the same genomic scaffold (Fig. S1). The DNA coding sequences of these three genes share over 90% sequence identity, suggesting that these three genes may have arisen from gene duplication events, which has been suggested as a general mode of sHSP evolution[51–54].

**sHSPs are highly expressed in tardigrades**. Two independent studies have conducted RNA-seq experiments to probe transcriptional changes in *H. exemplaris* during desiccation[48,49]. Here, we re-analyzed these data, mapping reads from both studies to a single version of the genome—the same version from which we identified the sHSPs[49]. In the Boothby et al. (2017) dataset, seven of the nine sHSPs were significantly upregulated in drying tardigrades at an FDR < 0.01 (Fig. 1b). In the Yoshida et al. (2017) dataset, only HSP23 was significantly upregulated in dried tardigrades (Fig. 1c). These differences might be explained by stage of drying (i.e., drying vs. dried), desiccation conditions, or sample collection methods used in these two studies. Regardless of the relevant differences, in each study, transcripts encoding several of the sHSPs were among the most highly abundant transcripts. We conclude that the transcripts encoding these sHSPs are highly abundant as animals experience desiccation.

**Several sHSPs can improve bacterial desiccation tolerance**. While abundance of sHSPs in desiccation tolerant organisms like nematodes, brine shrimp, and now tardigrades suggests that sHSPs may help proteins resist desiccation, we were motivated to test this hypothesis directly. To first determine if tardigrade sHSPs can promote desiccation tolerance, we heterologously expressed each sHSP in BL21 *E. coli*. Each construct was well-expressed in cells (Fig. S2a). However, HSP23 and HSP25, along with a truncated GFP control, had limited solubility in bacteria (Fig. S2b). We desiccated bacteria overexpressing each sHSP. Expression of HSP21, HSP24.6, HSP25.1, and HSP38 significantly improved desiccation survival relative to GFP-expressing control bacteria (Fig. 2). Bacteria expressing each of these four proteins exhibited over 100-fold improvement in desiccation survival relative to GFP-expressing control bacteria. Thus, at least some sHSPs are sufficient to improve desiccation tolerance. The limited solubility of HSP23 and HSP25 likely limit their ability to protect bacteria during desiccation, so it remains unclear if they would be protective if soluble. Other sHSPs like HSP17, HSP19, and HSP20 did not improve bacterial survival even though they were soluble, suggesting that they lack some property that is shared amongst at least HSP21, HSP23, HSP25.1, and HSP38.

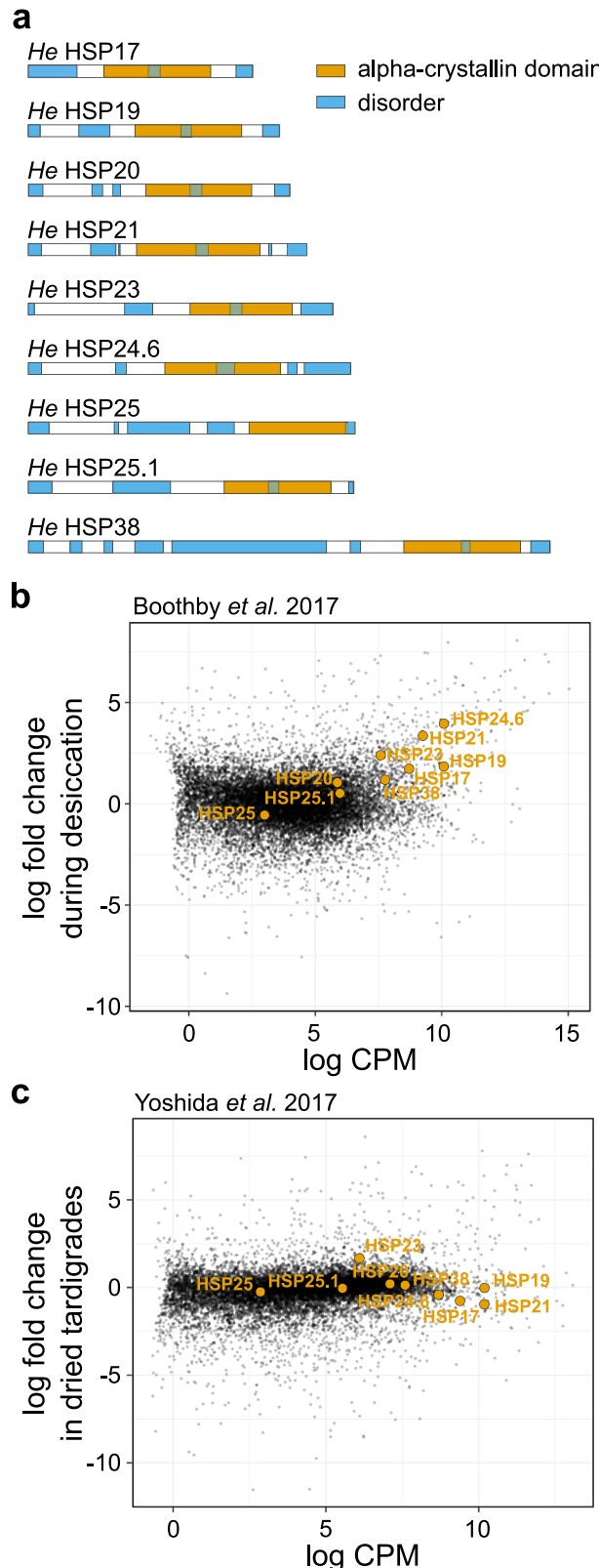

**Fig. 1 The tardigrade _Hypsibius exemplaris_ has nine small heat shock proteins, several of which are highly expressed. a** Nine sHSPs were identified that conform to the typical structure containing an alpha-crystallin domain flanked by disordered N-terminal and C-terminal domains. **b** Seven of nine sHSPs were upregulated during desiccation in RNA-seq data from Boothby et al. 2017 (HSP17, HSP19, HSP20, HSP21, HSP23, HSP24.6, and HSP38, FDR < 0.01). sHSP transcripts are highlighted on a plot of expression fold-change compared to relative abundance from mRNA-seq. **c** Only one sHSP (HSP23) was significantly upregulated during desiccation (FDR < 0.01) in RNA-seq data from Yoshida et al. 2017.

Therefore, we prioritized these two proteins for further characterization. To purify recombinant HSP21 and HSP24.6 for in vitro biochemistry, we added a 6xHis::TEV tag to each protein, purified the proteins with NiNTA columns, and cleaved the His tags with TEV protease (Fig. S2c, d).

Many sHSPs from different species are reported to form oligomeric complexes[17,55–58]. For example, p26 of Artemia forms a roughly 700 kDa multimeric assembly of ~27 monomers[40]. To determine if HSP21 and HSP24.6 form similar oligomeric assemblies, we visualized protein by negative staining TEM and determined the size distribution of protein complexes by mass photometry. Negative staining TEM of HSP21 revealed the presence of large structures that had some variability in size and shape (Fig. 3a). Mass photometry analysis showed a population of protein at low molecular weight that could represent monomers or small oligomeric complexes, and a second population of protein spanning a wide range of masses (Fig. 3b). This latter, broad peak likely represents the complexes seen with TEM. In contrast to HSP21, HSP24.6 formed slightly more homogenously-sized structures approximately 15–20 nm in diameter (Fig. 3c). By mass photometry we again detected peaks indicative of two populations of mass species. In this case, the peak representing larger multimeric complexes was more pronounced. Given the breadth of the peaks and range of masses included, we consider it likely that oligomeric complexes of variable numbers of monomers are represented. Assembly of higher-order complexes does not appear to be sensitive to particular buffers. Visualization of protein diluted in water (Fig. 3a, c), yielded similar results to protein diluted in TEN buffer (Fig. S3). Formation of HSP21 and HSP24.6 multimeric assemblies is consistent with reports from other sHSPs[14,59], and suggests that they may therefore harbor similar biochemical functions as chaperones.

**sHSPs can improve bacterial heat shock survival and chaperone against heat-induced protein aggregation.** The canonical chaperone function of sHSPs is to limit heat-induced protein aggregation. Citrate synthase (CS) is a common model enzyme used for studies of protein aggregation[3,60,61]. We tested if HSP21 or HSP24.6 could limit temperature-induced aggregation of CS. Heating at 43 °C induced aggregation of the enzyme that could be reduced by the addition of either HSP21 or HSP24.6 (Fig. 4a). HSP24.6 supplementation resulted in a more significant reduction of protein aggregation. In contrast, the addition of bovine serum albumin (BSA), used as a control, at the same molar ratio did not reduce CS aggregation. At higher concentrations, each sHSP could further limit thermal aggregation of CS, although BSA also had an effect at this concentration and was indistinguishable from HSP21 (Fig. S4). These data indicate that HSP24.6 is more effective at chaperoning against heat-induced aggregation.

To test if HSP21 and HSP24.6 could improve bacterial heat shock survival, we exposed BL21 _E. coli_ overexpressing these proteins to heat stress. We chose a stress of 52 °C for 1 h as a

**HSP21 and HSP24.6 can form higher-order complexes**. We were curious about the biochemical properties of sHSPs that could allow them to promote desiccation tolerance. HSP21 and HSP24.6 were the two proteins that conferred the largest improvement of bacterial desiccation survival (Fig. 2). Intriguingly, these were also the two genes that were most upregulated by desiccation in the Boothby et al. 2017 dataset (Fig. 1b).

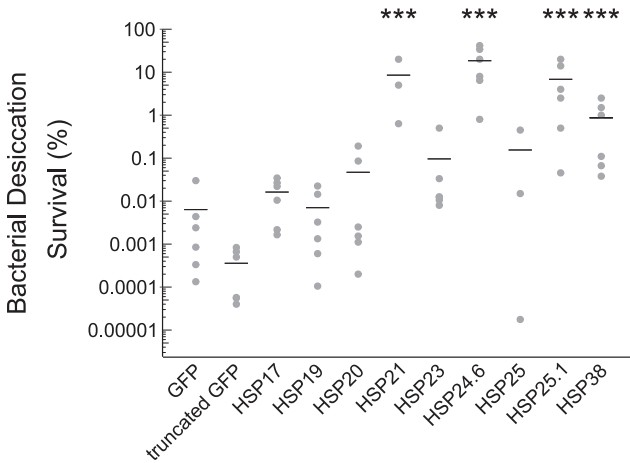

**Fig. 2 Tardigrade sHSPs can improve bacterial desiccation tolerance.** Desiccation survival of BL21 *E. coli* was significantly different across conditions ($p < 0.001$, 1-way ANOVA, $n = 3$–6). Bacterial survival was significantly improved relative to GFP-expressing bacteria by heterologous expression of HSP21 ($p < 0.001$), HSP24.6 ($p < 0.001$), HSP25.1 ($p < 0.001$), or HSP38 ($p < 0.001$). Dunnett's post-hoc test was used to compare each condition to GFP-expressing controls. Individual data points represent independent replicates and lines depict mean survival. ***$p < 0.001$.

condition for heat stress based on the survival of GFP-expressing bacteria across a range of temperatures (Fig. S5a). Heterologous expression of HSP24.6, but not HSP21, improved bacterial heat shock survival (Fig. 4b); perhaps surprisingly, none of the other tardigrade sHSPs significantly improved bacterial heat shock survival (Fig. S5b).

We hypothesized that HSP24.6 improved bacterial heat shock survival by chaperoning against protein aggregation. To test this hypothesis, we exposed total soluble protein lysate from the same strain of bacteria to heat stress. Indeed, we observed significant aggregation of the bacterial proteome when measuring light scattering ($A_{340}$) or by separating soluble (supernatant, S) and insoluble (pellet, P) fractions by centrifugation and visualizing these fractions with SDS-PAGE (Fig. 4c, d). Supplementation with a 1:1 mass ratio of HSP21 led to a modest reduction in protein aggregation, and the addition of HSP24.6 markedly reduced protein aggregation (Fig. 4c, d). We did not observe a significant effect of BSA in limiting protein aggregation. These results suggest that HSP21 has modest chaperone activity to limit thermal aggregation of proteins and that HSP24.6 is highly effective at chaperoning against protein aggregation at high temperature. Total soluble protein lysate from bacteria also contains a wide variety of proteins with unique biophysical properties. SDS-PAGE analysis of soluble and insoluble protein shows that HSP21 and HSP24.6 can promote the solubility of a broad range of proteins (Fig. 4d), indicating that the chaperone function of these sHSPs is likely general rather than specific to particular clients.

**HSP21 and HSP24.6 can limit desiccation-induced protein aggregation.** We were specifically interested in the possibility that tardigrade sHSPs may be able to function in the context of desiccation by acting as molecular shields. We tested the ability of HSP21 and HSP24.6 to limit desiccation-induced aggregation of citrate synthase. We purified a third tardigrade sHSP, HSP17 (Fig. S2e), that was not sufficient to improve bacterial desiccation tolerance (Fig. 2), with the expectation that this protein may serve as a negative control. We also included lysozyme as a negative control that we did not expect to impact the aggregation of CS.

BSA was included as a positive control with known potency as an excipient during desiccation[62,63]. As further evidence for a protective role of BSA, we found that it was sufficient to improve bacterial desiccation survival, despite low levels of expression (Fig. S6). Multiple rounds of desiccation led to increasing aggregation of CS, and a range of concentrations of sHSPs and BSA, could alter the extent of aggregation (Fig. 5a–e)[3]. None of the experimental protectant proteins when desiccated alone demonstrated significant aggregation as measured by light scattering ($A_{340}$) (Fig. S7). The addition of HSP21 reduced the aggregation of CS after one round of desiccation and rehydration ($p = 0.002$, 1-way ANOVA, $n = 4$–9). HSP24.6 significantly reduced aggregation of CS after both one ($p < 0.001$, 1-way ANOVA, $n = 4$–9) and two rounds of desiccation ($p = 0.003$, 1-way ANOVA, $n = 4$–9, Fig. 5f). Addition of BSA also protected CS from desiccation-induced aggregation ($p < 0.001$, 1-way ANOVA, $n = 4$–9). In contrast, supplementation with HSP17 caused an increase in aggregation ($p = 0.03$, 1-way ANOVA, $n = 3$–9). Interestingly, there were minimal differences in CS solubility across experimental conditions after multiple rounds of desiccation (Fig. S8). It is possible that some additives change the size or density of aggregates without shifting the overall balance of CS solubility. For example, the formation of smaller aggregates in the presence of some protectants could result in less light scattering while not impacting the solubility of total protein.

Ultimately, protein misfolding and aggregation can result in loss of protein function. To more directly test the protective ability of sHSPs during desiccation, we measured CS activity after four rounds of desiccation and rehydration. At a 2:1 molar ratio, HSP17, HSP21, and HSP24.6 each prevented the desiccation-induced loss of CS activity (Fig. 5g). Lower concentrations of HSP21 (1:1 molar ratio) and HSP24.6 (1:5 molar ratio) were sufficient to limit the loss of CS activity. At a concentration of 10 µM HSP21 and HSP24.6 functioned similarly to BSA (despite, with a molecular weight of 66 kDa, an equimolar amount of BSA representing approximately 3× the mass of sHSP monomers). It is interesting that HSP17 could limit the loss of CS activity despite an apparent increase in aggregation after two rounds of desiccation. Several sHSPs from yeast, *C. elegans*, and *E. coli* have been shown to exert cytoprotective functions by promoting the sequestration of misfolded proteins in inclusions[64]. It is possible that HSP17 acts similarly to sequester CS and protect it from irreversible denaturation. Overall, we conclude that HSP21 and HSP24.6 can limit desiccation-induced aggregation of CS and can allow for the retention of enzymatic function.

## Discussion

Small heat shock proteins are well-studied chaperones that are upregulated upon several stress conditions, and mutations of sHSPs has been associated with human disease[25,65]. We were initially intrigued by reports of the high abundance of sHSPs in some desiccation-tolerant organisms like *Artemia* and *C. elegans*[33,34,37]. We found that in two independent RNA-seq datasets of *H. exemplaris*, sHSPs were among the most abundant transcripts detected (Fig. 1b, c). The Boothby et al. dataset showed significant induction of many sHSPs while the Yoshida et al. dataset did not. Another study assessed expression of two sHSPs from the tardigrade *Milnesium tardigradum* during heat shock and anhydrobiosis, with one sHSP being significantly upregulated by thermal stress, but neither by anhydrobiosis[66]. It is possible that sHSPs are more transcriptionally responsive to temperature fluctuations, which could explain differences in induction between the two datasets if there were differences in temperature when tardigrades were collected. It is also likely that sHSPs are variably expressed during different stages of drying and

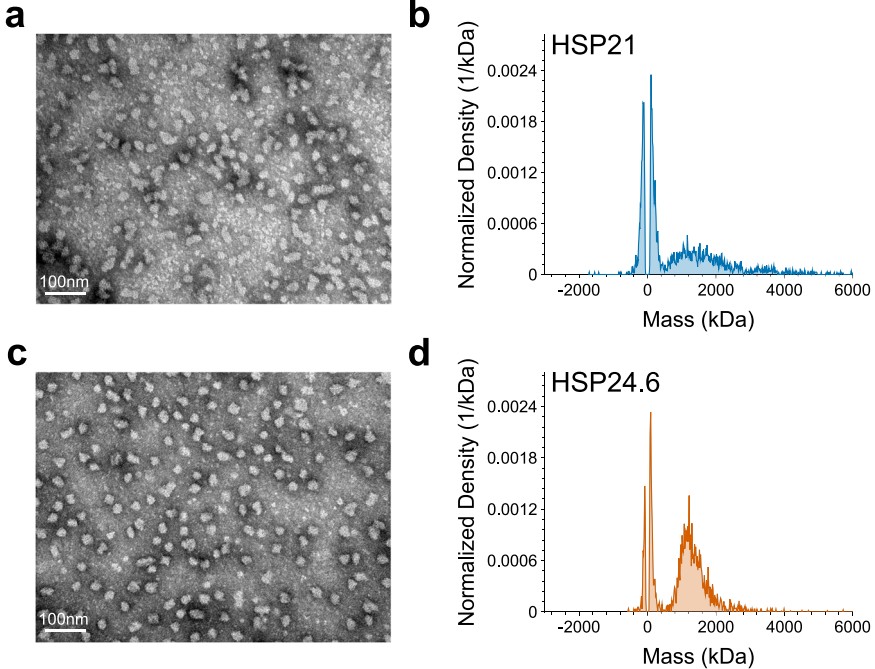

**Fig. 3 HSP21 and HSP24.6 can form large oligomeric complexes. a** Negative staining TEM of HSP21 reveals large assemblies of variable size and shape. **b** Mass photometry analysis indicates two populations of HSP21, a peak of smaller mass species—likely comprising limited oligomeric assembly—and a broad peak of larger mass species that likely includes a range of oligomeric states. **c** Negative staining TEM of HSP24.6 shows particles of more regular size and shape than HSP21. **d** Mass photometry analysis reveals HSP24.6 protein complexes of larger size that likely represent large oligomeric assemblies, in addition to a peak at lower mass representing smaller assemblies. Proteins were diluted in 1x PBS for mass photometry, and in molecular grade water for TEM.

particularly sensitive to the time of sampling during the process of tun formation and preparation for anhydrobiosis. Besides differences in induction, RNA-seq data show the relative abundance of sHSP transcripts even in unstressed tardigrades. Transcriptional induction may indicate an adaptive response, but is not a necessary condition for function if proteins are constitutively present. Therefore, this result suggests that perhaps these proteins are constitutively present and can stabilize the proteome under normal conditions in addition to during stresses like heat or desiccation, similar to what has been proposed for sHSPs in other organisms, including *C. elegans* and mammals, where selected sHSPs have been shown to participate in development and differentiation[67,68].

Although tardigrades are renowned for their desiccation survival, they are not noted for their survival of heat stress unless in the tun state[69,70]. This raises the possibility that some tardigrade sHSPs may have evolved properties that enable them to function during desiccation, perhaps at the expense of thermal tolerance. Given the typical association with sHSPs and heat stress, we were somewhat surprised to find that HSP21, HSP24.6, HSP25.1, and HSP38 could each improve bacterial desiccation survival (Fig. 2), but only HSP24.6 was sufficient to improve bacterial heat shock survival (Fig. S5b). Notably, not all sHSPs function during heat shock, and some act as chaperones instead during cold stress or other conditions[71]. It is possible that some tardigrade sHSPs have, in fact, undergone selection for function during desiccation. However, using bacteria as a system for heterologous expression makes these differences challenging to interpret. A clear comparison of sHSP function during stresses is confounded by variables like the levels of protein expression or solubility, conditions for bacterial heat stress and desiccation that may not be optimal for the function of tardigrade proteins, or other caveats of putting eukaryotic proteins in prokaryotic cells. Nonetheless, heterologous expression experiments like these are a powerful and

efficient way to test for protectants that are sufficient to promote survival and generalizable beyond the specific biology of a tardigrade. With an eye towards developing better protectants to stabilize biomaterials or produce drought-tolerant crops, these are the types of protectants that are of greatest interest. If tardigrade proteins are uniquely suited as desicco-protectants, it is intriguing to consider the possibility for such applications.

To gain some insight into whether sHSPs from tardigrades may be uniquely optimized for conditions of desiccation, we tested if human sHSPs could provide similar improvement to bacterial desiccation survival. Heterologous expression of five of the ten human sHSPs (HSPB1/Hsp27, HSPB4/alpha A crystallin, HSPB5/alpha B crystallin, HSPB7/cvHSP, and HSPB10/ODF1) improved bacterial desiccation tolerance (Fig. S9). Again, high expression levels and solubility may explain some of the ability of particular sHSPs like HSPB1, HSPB4, and HSPB5 to improve bacterial desiccation survival (Fig. S9b, c). These results suggest that tardigrade sHSPs may not harbor particularly unique chaperone properties for desiccation, but rather, that animal sHSPs in general might harbor potency to contribute to desiccation tolerance. Thus, desicco-protection is emerging as a conserved property of some sHSPs across different species. So why are human cells still desiccation-sensitive if some human sHSPs can also promote bacterial desiccation tolerance? Desiccation is a harsh stress that causes widespread cellular damage (for example to DNA, RNA, and membranes) that precludes survival even if stress to the proteome is lessened. Human sHSPs may limit protein aggregation and improve survival in other, less severe, contexts like osmotic stress that mimic the depletion of subcellular water and the concentration of cytosolic components. It is further possible that in human cells sHSPs may not be present in appropriate quantities or subcellular locations required for function, or that other protectants must also be provided. For example, heterologous expression of the sHSP p26 from brine shrimp did not

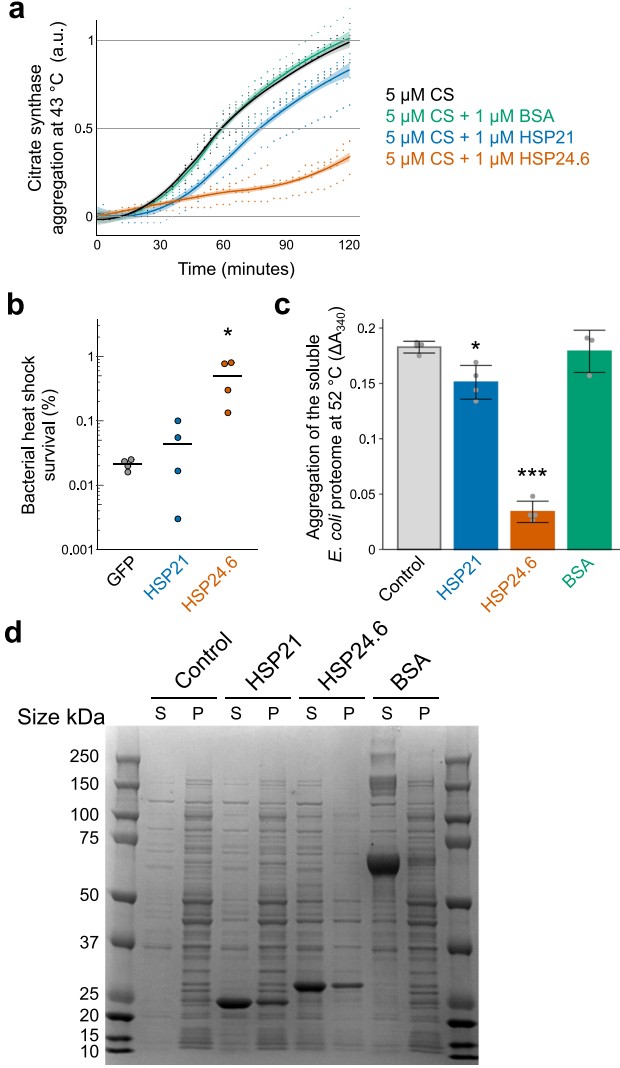

**Fig. 4 Tardigrade sHSPs can chaperone against heat-induced protein aggregation. a** HSP21 and HSP24.6 limit heat-induced aggregation of citrate synthase at 43 °C. Aggregation is plotted as the change in $A_{340}$ relative to that of 5 μM citrate synthase alone at 2 h. Datapoints from four biological replicates are shown along with a fitted line and 95% confidence interval. **b**) Heterologous expression of HSP24.6 improved heat shock survival of BL21 *E. coli* ($p = 0.02$, Dunnett's test, $n = 4$). **c** HSP21 and HSP24.6 limited heat-induced aggregation of the water soluble BL21 *E. coli* proteome. In vitro aggregation of soluble protein following 1 h at 52 °C was significantly different across conditions shown ($p < 0.001$, 1-way ANOVA, $n = 3–4$). HSP21 ($p = 0.01$, Dunnett's test, $n = 4$) and HSP24.6 ($p < 0.001$, Dunnett's test, $n = 4$) reduced aggregation, while BSA ($p = 0.96$, $n = 3$, Dunnett's test) did not have an effect. sHSPs or BSA were added at a 1:1 mass ratio with *E. coli* protein. **d**) Soluble (supernatant, S) and insoluble (pellet, P) protein fractions were assessed by SDS-PAGE. A representative Coomassie-stained gel shows soluble and insoluble protein following heat shock at 52 °C for 1 h. The pelleted insoluble fraction was concentrated 4× before loading. *$p < 0.05$, ***$p < 0.001$.

improve the viability of drying 293H cells unless trehalose was also present[41].

We found that HSP21 and HSP24.6 form oligomeric complexes (Fig. 3), similar to what has been described for other sHSPs[18,40,55–58]. In some cases, the ability of sHSPs to oligomerize has been shown to be essential for their chaperone properties, and some of the human sHSPs that could protect bacteria from desiccation are also known to oligomerize[72–75]. We

speculate that oligomerization of HSP21 and HSP24.6 may be required for their activity in limiting heat- and desiccation-induced protein aggregation. However, further work is required to test this hypothesis directly. The dynamic equilibrium of assembly and disassembly of sHSP oligomeric complexes is often sensitive to environmental factors like temperature or pH[19], which raises the possibility that these, or other factors like osmolyte concentration, could alter the assembly of sHSP complexes during drying. It is also possible that these sHSPs may hetero-oligomerize in the presence of other sHSPs to form alternative functional assemblies[76,77]. Future analysis of native sHSP assemblies may inform the extent to which such interactions exist and are important in the context of desiccation.

In conclusion, we define a family of sHSPs from the tardigrade *H. exemplaris* and present evidence for their involvement in the desiccation response. Many sHSP transcripts were present in significant quantities in tardigrades, and overexpression of several of these proteins could improve bacterial desiccation survival. HSP24.6 was particularly effective at promoting desiccation survival, likely by limiting protein aggregation. HSP24.6 also improved bacterial heat shock survival and limited heat-induced protein aggregation. Thus, HSP24.6 is a general chaperone that may function to maintain proteostasis in varying stress conditions. It is possible that sHSPs may have other functions beyond limiting protein aggregation that could contribute to desiccation survival. sHSPs have been shown in some contexts to impact diverse cellular components and processes such as protein degradation, membrane fluidity, cytoskeletal organization, and apoptosis[24,78–83]. sHSPs contain intrinsically disordered regions and can undergo liquid-liquid phase separation, forming or being recruited inside biomolecular condensates[68,84–87]. Biomolecular condensates are typically enriched for aggregation-prone disordered proteins, and mammalian sHSPs were shown to prevent condensate conversion from a dynamic liquid-like state into an irreversibly aggregated state, with important implications for cellular stress responses and human disease[85–87]. Whether tardigrade sHSPs might also help to maintain reversible phase-transitions and whether this may contribute to desiccation survival, similar to what has been suggested for LEA proteins[88], is still unknown. It will be of further interest to determine the properties that allow some sHSPs to function during desiccation and to explore the endogenous roles of sHSPs in tardigrades.

## Methods

**Identification and cloning of *H. exemplaris* sHSPs.** Small heat shock proteins of *Hypsibius exemplaris* were identified by BLAST using the ten human HSPB sequences and 12 *Drosophila* sHSP sequences as BLAST queries against the *H. exemplaris* transcriptome version 3.1.5[49]. Protein sequences from top hits were surveyed for alpha crystallin domains and regions of disorder, characteristics of conserved sHSPs. These features were annotated based on identification of the alpha crystallin domain from NCBI conserved domain searches and regions of predicted disorder annotated by prDOS[89,90]. Protein sequences of sHSPs are included in Supplementary Table 2.

sHSPs were cloned into pDest17 for bacterial expression. Tardigrade (*Hypsibius exemplaris* strain Z151) RNA was isolated by established methods and converted to cDNA using the SuperScript III First-Strand Synthesis System (Invitrogen, 18080051)[91]. Primers were designed to amplify transcripts from total cDNA, and coding sequences were amplified and assembled into linearized pDest17 using NEBuilder Hifi Assembly 2× master mix (NEB, E2621). NEB 5-alpha *E. coli* were transformed with the assembly product, and individual colonies were grown overnight and miniprepped. The gene insert regions were sequenced to confirm that the coding region matched that of the genome (3.1.5) and were correctly assembled into the vector. Synonymous mutations in coding sequences were present in some cases. BV898_14401 (HSP20) was unable to be cloned from cDNA so was instead cloned into the expression vector from a synthesized gBlock (Integrated DNA Technologies).

**Analysis of desiccation-induced expression.** Two previous studies conducted RNA-seq on hydrated and desiccated tardigrades[48,49]. Here, we have re-analyzed those datasets by mapping reads to the most recent version of the genome (v3.1.5).

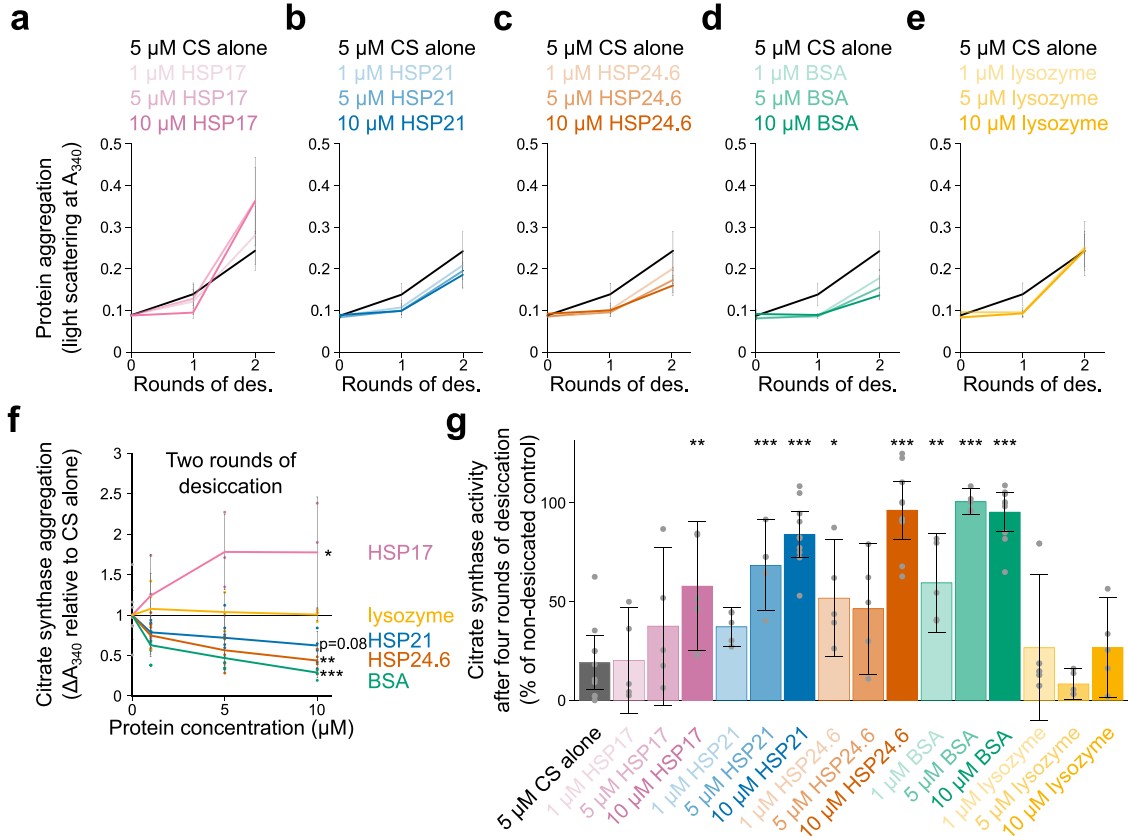

**Fig. 5 HSP21 and HSP24.6 limit desiccation-induced protein aggregation.** Light scattering ($A_{340}$) values indicate aggregation of 5 μM citrate synthase over two rounds of desiccation and rehydration and the effects of adding HSP17 (**a**), HSP21 (**b**), HSP24.6 (**c**), BSA (**d**), or lysozyme (**e**). **f** After two rounds of desiccation and rehydration, the change in $A_{340}$ across a range of concentration of additives showed an effect of concentration on aggregation for HSP17 ($p = 0.03$, 1-way ANOVA, $n = 3$–9), HSP24.6 ($p = 0.003$, 1-way ANOVA, $n = 4$–9), and BSA ($p < 0.001$, 1-way ANOVA, $n = 4$–9). **g** Citrate synthase was protected from desiccation-induced loss of function when supplemented with 10 μM HSP17 ($p = 0.009$), 5 μM HSP21 ($p < 0.001$), 10 μM HSP21 ($p < 0.001$), 1 μM HSP24.6 ($p = 0.05$), 10 μM HSP24.6 ($p < 0.001$), 1 μM BSA ($p = 0.006$), 5 μM BSA ($p < 0.001$), or 10 μM BSA ($p < 0.001$). Enzyme activity is plotted as a percentage of non-desiccated citrate synthase. *P*-values resulted from Dunnett's tests comparing to desiccated CS alone, $n = 5$–10. Mean values and standard deviation are plotted in (**a**–**g**). *$p < 0.05$, **$p < 0.01$, ***$p < 0.001$.

Reads were downloaded from the NCBI SRA database (SRP098585: GSM2472501 through GSM2472506 and PRJNA: SRX2663153, SRX2663154, SRX2527798, SRX2661843, SRX2661844, SRX2527616). Reads were mapped using Bowtie2 and counts were assigned with Featurecounts and using the annotation file associated with the genome version 3.1.5. Reads were aggregated at the level of genes. Only genes with more than one count in at least two samples were kept for differential expression analysis. Transcript abundance, fold changes, and FDR values were determined with EdgeR[92].

**Bacterial desiccation and heat shock survival**. Individual sHSPs were expressed in *E. coli* BL21 AI (Invitrogen, C607003) to determine if heterologous expression could confer desiccation tolerance. BL21 AI cells were transformed with the sequence-verified expression plasmids. Bacteria were grown overnight in 5 mL LB with Ampicillin and diluted 1:20 into LB with Ampicillin and 0.2% L-arabinose to induce expression. Induction cultures were grown for 4 h at 37 °C with shaking. OD600 of induced bacteria was measured and the densities of bacteria were normalized. A dilution series of bacteria was plated to determine the control cfu. Bacteria were pelleted, supernatant removed, and placed in a speedvac for overnight desiccation. Bacteria were then rehydrated with the same volume in which they were initially suspended and a dilution series was plated. Survival was calculated as the cfu after desiccation divided by the control cfu. For heat shock experiments, bacteria were prepared as for desiccation and heated for 1 h in a Thermocycler. Non-heated samples were kept as controls. Serial dilutions were plated and survival calculated as the ratio of cfu from heated samples to controls.

**Protein purification**. To purify HSP17, HSP21, and HSP24.6, a 6× His epitope tag and TEV cleavage site were cloned onto the N-terminus of the coding sequence of each gene in expression plasmids. Expression vectors were sequence-verified and transformed into BL21 AI *E. coli* (Invitrogen, C607003). Overnight cultures of bacteria in LB with Ampicillin were inoculated at a 1:20 ratio into 1–2 L of LB with Ampicillin and 0.2% L-arabinose for induction. Cultures were induced for 6 h at

37 °C with shaking. Bacteria were harvested by centrifugation at 5000 rpm for ten minutes and concentrated into pellets.

Bacterial pellets were resuspended in NiNTA binding/wash buffer A (20 mM sodium phosphate buffer, 500 mM NaCl, 20 mM imidazole, pH 7.4) with lysozyme (Sigma, L6876), DNAse I (Thermo, 18047019), and a protease inhibitor cocktail (Fisher, PIA32965) and sonicated on ice. Lysates were spun at $14,000 \times g$ for 1 h to clear cellular debris and insoluble protein. Soluble and insoluble fractions of cell lysate were run on gels to ensure that protein to be purified was present in the soluble fractions. Lysates were filtered (0.45 μm filter) and loaded onto a HisTrap column (Cytiva) with 1 mL binding capacity. The column was washed with NiNTA wash buffer (with 20 mM imidazole) and eluted with the same buffer with 250 mM imidazole. The column was then flushed with buffer with 500 mM imidazole.

Protein eluted from the column was treated with TEV protease to cleave the 6× His tag. The protein was dialyzed into HBS for the TEV digest. TEV protease was added at a mass ratio of 25:1 (target protein:protease) and allowed to proceed overnight at 4 °C. The digest was dialyzed back into NiNTA buffer A and passed back over the regenerated, re-equilibrated column. The flowthrough was collected as released target. Uncleaved protein was then eluted with 250 mM imidazole. The cleaved target protein was dialyzed into PBS, aliquoted, and flash frozen in liquid nitrogen. Aliquots were stored at −80 °C.

**Negative staining TEM**. Protein samples were visualized by negative-stain transmission electron microscopy. Concentrations were 100 μg/mL of protein for samples diluted in water and 50 μg/mL of protein diluted in TEN buffer. A glow-discharged formvar/carbon-coated 400 mesh copper grid (Ted Pella, Inc., Redding, CA) was floated on a 20 μl droplet of the sample suspension for 10 min, transferred quickly to 2 drops of deionized water followed by a droplet of 2% aqueous uranyl acetate stain for 1 min. The grid was blotted with filter paper and air-dried. Samples were observed using a JEOL JEM1230 transmission electron microscope operating at 80 kV (JEOL USA INC., Peabody, MA) and images were taken using a Gatan

Orius SC1000 CCD camera with Gatan Microscopy Suite version 3.10.1002.0 software (Gatan, Inc., Pleasanton, CA).

**Mass photometry**. A Refeyn One mass photometer was used to assess the distribution of mass species of purified sHSPs[93,94]. Coverslips were prepared by sonication in isopropanol for five minutes, washed in water twice, sonicated for 5 minutes in water, and washed once more in water. Coverslips were dried before applying a sample well cassette. 10 μL of PBS was added to a well and the focus was adjusted. 10 μL of diluted sHSPs were added to each well for final concentrations of 500 nM (HSP21), or 1 μM (HSP24.6). Testing a range of concentrations provided similar results to those reported. The raw contrast values were converted to mass values by normalizing to a mass calibration with NativeMark unstained protein ladder (Thermo, LC0725).

**Chaperone assays**. To assess the chaperone activity of sHSPs, the aggregation-prone model enzyme citrate synthase was used. 5 μM citrate synthase (Sigma, C3260) was incubated at 43 °C in a BioTek Synergy H1 plate reader with orbital shaking. sHSPs or bovine serum albumin (BSA) were added at a concentration of 1 μM or 5 μM. Absorbance at 340 nm was read at 3 min intervals. To test for aggregation of total E. coli protein, soluble lysate was collected from BL21 AI cells transformed with pUC19. Cells were resuspended in water and sonicated on ice. The crude lysate was spun twice at 14,000 rpm for ten minutes at 4 °C, and the soluble supernatant was retained. Protein was quantified with the BioRad Protein Assay kit. Lysates were set up with 50 μg of the soluble lysate and 50 μg of either sHSP protein or BSA. Protein mixtures were heated at 52 °C for 1 h. The absorbance at 340 nm was read before and after heating. To visualize soluble and insoluble protein, heated samples were centrifuged at 14,000 rpm for ten minutes at 4 °C. The soluble fraction (100 μL) was moved to a new tube. Pelleted insoluble debris was resuspended in 50 μL of sample buffer. These fractions were run on a 4–12% BT gel with 10 μL of soluble protein added to 10 μL of 2× sample buffer and 20 μL of insoluble protein in sample buffer. This is effectively a 4× concentration of the insoluble fraction.

To test for desiccation-induced protein aggregation, 5 μM citrate synthase solutions were prepared in 0.1× PBS and supplemented with varying concentrations of HSP17, HSP21, HSP24.6, BSA, or lysozyme. Solutions were subjected to multiple 2 hr rounds of desiccation in a savant speedvac concentrator and rehydrated in molecular grade water. Light scattering (A340) was measured before desiccation and after successive rounds of desiccation and rehydration.

To determine solubility of CS after two and four rounds of desiccation, rehydrated protein was spun at 14,000 rpm for ten minutes at 4 °C. Soluble supernatant was moved to a new tube and the pelleted insoluble fraction was resuspended in 50 μL of sample buffer and 50 μL of molecular grade water. Soluble and insoluble fractions were run on a 4–12% BT gel with 8 μL of soluble protein added to 8 μL of 2× sample buffer and 16 μL of insoluble protein in sample buffer.

**Citrate synthase enzyme activity**. A Citrate Synthase Activity Assay Kit from Sigma-Aldrich (MAK193) was used to measure the enzyme activity of citrate synthase before and after desiccation. Because significant enzyme function was retained with two rounds of desiccation, we used four rounds of desiccation as the assay endpoint. Samples were diluted 1:100 to monitor activity. Absorbance at 512 nm was read every five minutes for one hr in a BioTek Synergy H1 plate reader. Activity was calculated according to the kit's instructions and normalized to the activity of CS from samples that were not desiccated.

**Statistics and reproducibility**. For gene expression analysis, statistics were calculated by EdgeR and transcripts were assigned an FDR value. For heterologous expression survival experiments in bacteria, a 1-way ANOVA was used to first test for any difference across conditions. If this was significant, a post hoc Dunnett's test was used to determine which conditions were significantly different from the GFP-expressing control strain. Similarly, for protein aggregation and enzyme activity experiments, 1-way ANOVAs followed by post hoc Dunnett's tests were employed to determine significant differences from a control. The number of independent biological replicates for each experiment is noted in Figure legends or the text.

**Reporting summary**. Further information on research design is available in the Nature Portfolio Reporting Summary linked to this article.

## Data availability

All data described in the manuscript are contained within the manuscript and supporting information. Raw data from previous RNA-seq experiments are available from the SRA database[48,49]. Source data for figures can be found in Supplementary Data 1.

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

## Acknowledgements

We thank Nathan Nicely of the Protein Expression and Purification core facility at UNC for purification of HSP17, HSP21, and HSP24.6 and for training on the mass photometer. We thank Kristen White and Kathleen A. Clardy of the Microscopy Services Laboratory for sample preparation and TEM imaging. We are grateful to Roy A. Quinlan and Muhammad Alansari for helpful discussions. This work was supported by the National Institutes of Health (F32GM131577, awarded to J.D.H.), the National Science Foundation (IOS 2028860, awarded to BG), and MIUR (Departments of excellence 2018-2022, E91I18001480001, awarded to S.C). The Protein Expression and Purification core facility at UNC is supported by the National Cancer Institute of the National Institutes of Health under award number P30CA016086. The Microscopy Services Laboratory, Department of Pathology and Laboratory Medicine, is supported in part by P30CA016086 Cancer Center Core Support Grant to the UNC Lineberger Comprehensive Cancer Center. The content is solely the responsibility of the authors and does not necessarily represent the official views of the National Institutes of Health.

## Author contributions

J.D.H.: Conceptualization, formal analysis, investigation, writing – original draft, writing – review and editing, visualization, project administration, funding acquisition. S.C.: Conceptualization, writing – review and editing, supervision, funding acquisition. B.G.: Conceptualization, writing – review and editing, supervision, funding acquisition.

## Competing interests

The authors declare no competing interests.
