## [Peer Review File · Communications Biology]

Reviewers' comments:

Reviewer #1 (Remarks to the Author):

Small heat shock proteins (sHsps) function as ATP-dependent chaperones to prevent the uncontrolled aggregation of misfolded proteins. Here the authors studied the function of sHsps from the tardigrade *Hypsibius exemplaris* and their roles in providing protection against desiccation. Various of these sHsps are highly expressed, yet reports on their increased expression during desiccation are contradictory. The authors tested the capacity of these sHsps in protecting cells from desiccation by heterologous expression in *E. coli*. Some of the tested sHsps strongly increased bacterial survival, yet the underlying molecular reasons remained unexplored. sHsps that showed activity *in vitro* were subsequently characterized *in vitro* revealing classical sHsp features and chaperone activities. Notably, the authors reconstitute the protective function of sHsps against desiccation *in vitro* by showing that sHsps maintain the model substrate Citrate Synthase in an active state.

The study includes interesting data and point to a novel physiological role of sHsps in protecting cells and proteins against desiccation. Why some sHsps show activity *in vivo* whereas others don't remained unexplored. Here, additional controls are recommended (see below) to address sHsp specificity. It also seems relevant to analyze the protective function of sHsps against desiccation in more detail (see below). These points will bolster the central point of the study and they should be included in a revised manuscript.

Major points:

Figure S2A: While all sHsps are expressed at similar levels in *E. coli* cells, it is not clear whether they accumulate in soluble or insoluble cell fractions. The authors need to determine the solubilities of the overproduced sHsps as potential differences might be sufficient to explain their diverse activities *in vivo*. Similarly, the authors need to provide expression levels and solubilities of overproduced human sHsps (FigureS5).

sHsps differ in their disordered N- and C-terminal extensions. The authors are asked to comment whether sHsps that protect *E. coli* cells against desiccation exhibit common and unique features in these regions separating them from inactive sHsps.

Figure 4A: The authors only test a single ratio between the substrate Citrate Synthase and the sHsps (and BSA control) during thermal denaturation. Since aggregation is only suppressed partially (e.g. by Hsp21) it is recommended to probe for activities of higher sHsp (and BSA) concentrations.

Figure 5: The authors monitor aggregation of Citrate Synthase (CS) by absorbance measurements. It is however unclear which fraction of the substrate is aggregating under the experimental conditions. This is highly relevant as Hsp21 shows only minor protective activity when monitoring CS aggregation by absorbance measurements (after 2 rounds of desiccation) but keeps the substrate in an active state after 4 rounds of desiccation. The authors are therefore asked to determine CS solubilities in absence and presence of sHsps (and BSA) after 2/4 rounds of desiccation. It also seems necessary to show how sHsps protect CS from inactivation. Do they form complexes with the substrate? If so, what is the size of the complexes? Such analysis is relevant as it needs to be documented whether the protective function of sHsps during desiccation relies on their classical chaperone activity.

Figure 5: The authors use BSA as positive control as it was reported to protect proteins against inactivation upon desiccation. The observation that the tested sHsps are similarly active raises the question of specificity as no true negative control (added protein showing no activity) is provided. It is therefore crucial to test for activities of non-related control proteins to exclude that the observed effects simply rely on the presence of high protein concentrations added externally. This will document the specific protective function of sHsps under the given experimental conditions.

Reviewer #3 (Remarks to the Author):

Hibman et al. reports the capabilities of small HSPs of a tardigrade in enhancing desiccation tolerance when expressed in *E. coli*, and for the protection of enzymatic activities of citrate synthase *in vitro*. This work confirms the previous reports such as Ma et al. 2005 *Cryobiology* that sHSPs of anhydrobiotic species confer tolerance to desiccation in heterologous systems. (In Ma et al it was mammalian cells and synergistically with trehalose). I especially found Figure S5 to be intriguing, that sHSPs from humans show comparable heterologous protection as tardigrade sHSPs.

My first impression of this work is that while each of the data presented seem interesting, logical connections between the figures seem lacking (the authors do discuss this issue in the Discussions). For example, Fig. 3 shows that HSP21 and HSP24.6 to form small assemblies, but no data is shown regarding the contribution or necessity of this assembly (or even whether this assembly can be seen *in vivo*) in desiccation tolerance. Is this observable in the citrate synthase protection experiment? or in *E. coli*? Would buffer conditions disrupting these oligomerization impair desiccation tolerance?

In Figure 4, only HSP24.6 show increased heat tolerance, and not HSP21, and authors discuss that this may be due to differentiated usage of sHSPs for heat tolerance and for desiccation. However, HSP24.6 always show better protection from desiccation (in *E. coli* Fig.2, *in vitro* Fig.5).

In Fig.5, BSA is used as a positive control for protection from protein aggregation under desiccated conditions. Does BSA then improve desiccation tolerance in *E. coli*? This is not shown in Fig. 2. The key argument of the paper, as the title stands, is the inference of lowered protein aggregation by sHSPs and its effect in desiccation tolerance. However, Fig. 5 only measures A340 readings, which is the NADH consumption by citrate synthase, so it is a measure of enzymatic activity, rather than aggregation. Aggregation can be a reason for the loss in enzymatic activity, but other factors (such as simple denaturation) could also be considered. Lowered protein aggregation in desiccated *E. coli* is not shown, and it is therefore very difficult for the readers to connect heterologous desiccation protection, protein assembly, and *in vitro* protection of enzymatic activity, from the presented data.

Human sHSP is only shown for the heterologous expression in *E. coli*. Do they not assemble into oligomeric complexes, and do they not protect citrate synthase from desiccation? If they do, these properties are likely common to sHSP class of proteins, and not to be signified as tardigrade-specific as in the current title and discussions. Moreover, there are different ways these sHSPs can contribute to desiccation tolerance besides protection from protein aggregation, including the protection of membranes (see for example, Sales et al. *BBA* 2000). As described above, I do not see convincing data in the current manuscript to rule out these alternative pathways of protection.

I suggest to mention more clearly that desiccation tolerance by artemia sHSP in mammalian cells (Ma et al. 2005 *Cryobiology*), increased heat tolerance by them (Liang and MacRae 1999 *Dev. Biol.*), and *in vitro* protection of cytrate synthase activity by them (Ping et al. 1997 *Eur. J. Biochem*) have already been reported. Current introduction states rather strongly that sHSPs of animals are not well explored, but very similar experiment to the current manuscript has already been reported for artemia p26. I still see a value in doing similar work for sHSPs from other anhydrobiotic species, namely a tardigrade in this work, but nevertheless the previous efforts should be appropriately noted.

We thank the reviewers for their helpful comments and suggestions. We have significantly revised the manuscript based on the reviews, adding the new experiments shown in Figure 5a,e,g, Figure S2b,e, Figure S4, Figure S6, Figure S7, Figure S8, and Figure S9b,c. We believe that addressing the reviewers' concerns has improved the manuscript. Please see below for responses to each of the points raised.

Reviewer #1 (Remarks to the Author):

Small heat shock proteins (sHsps) function as ATP-dependent chaperones to prevent the uncontrolled aggregation of misfolded proteins. Here the authors studied the function of sHsps from the tardigrade *Hypsibius exemplaris* and their roles in providing protection against desiccation. Various of these sHsps are highly expressed, yet reports on their increased expression during desiccation are contradictory. The authors tested the capacity of these sHsps in protecting cells from desiccation by heterologous expression in *E. coli*. Some of the tested sHsps strongly increased bacterial survival, yet the underlying molecular reasons remained unexplored. sHsps that showed activity *in vitro* were subsequently characterized *in vitro* revealing classical sHsp features and chaperone activities. Notably, the authors reconstitute the protective function of sHsps against desiccation *in vitro* by showing that sHsps maintain the model substrate Citrate Synthase in an active state.

The study includes interesting data and point to a novel physiological role of sHsps in protecting cells and proteins against desiccation. Why some sHsps show activity *in vivo* whereas others don't remained unexplored. Here, additional controls are recommended (see below) to address sHsp specificity. It also seems relevant to analyze the protective function of sHsps against desiccation in more detail (see below). These points will bolster the central point of the study and they should be included in a revised manuscript.

Major points:

Figure S2A: While all sHsps are expressed at similar levels in *E. coli* cells, it is not clear whether they accumulate in soluble or insoluble cell fractions. The authors need to determine the solubilities of the overproduced sHsps as potential differences might be sufficient to explain their diverse activities *in vivo*. Similarly, the authors need to provide expression levels and solubilities of overproduced human sHsps (FigureS5).

Thank you for this suggestion. We now include protein gels showing solubility of tardigrade proteins expressed in bacteria (Figure S2b). We also now provide expression levels and solubilities of human sHSPs (Figure S9b,c). Limited solubility of some sHSPs may explain some of the effects on survival as we now discuss (Lines 162-166, 359-361). For this reason, we focus our conclusions on only those sHSPs for which we have positive evidence for their capacity to promote desiccation tolerance, and we take care to make no conclusions about proteins that were insoluble.

sHsps differ in their disordered N- and C-terminal extensions. The authors are asked to comment whether sHsps that protect *E. coli* cells against desiccation exhibit common and unique features in these regions separating them from inactive sHsps.

We have now tested the solubility of each sHSP (Figure S2b). Based on the results, we believe this could feasibly be the most salient property that impacted the ability of certain sHSPs to promote *E. coli* desiccation survival in our experiments. We agree with the reviewer that this merits discussing, so we have added discussion of this into the Results and Discussion sections (Lines 162-166, Lines 345-348). We did not identify any conserved sequence-specific features among tardigrade and human sHSPs that could improve desiccation survival. Given that the

disordered regions of the proteins are generally not well conserved, it is difficult to rule out possible structural homology that could be important for sHSP functions.

Figure 4A: The authors only test a single ratio between the substrate Citrate Synthase and the sHsps (and BSA control) during thermal denaturation. Since aggregation is only suppressed partially (e.g. by Hsp21) it is recommended to probe for activities of higher sHsp (and BSA) concentrations.

In response to the reviewer's concern, we have conducted these experiments. We now include new data showing that with higher concentrations of protectants (5 μ M), thermal aggregation of citrate synthase can be further limited (Figure S4). A molar equivalent ratio of the larger protein BSA (i.e. a larger total mass of protein) also had an effect at this higher concentration, though its effect was weaker than HSP24.6's (Figure S4).

Figure 5: The authors monitor aggregation of Citrate Synthase (CS) by absorbance measurements. It is however unclear which fraction of the substrate is aggregating under the experimental conditions. This is highly relevant as Hsp21 shows only minor protective activity when monitoring CS aggregation by absorbance measurements (after 2 rounds of desiccation) but keeps the substrate in an active state after 4 rounds of desiccation. The authors are therefore asked to determine CS solubilities in absence and presence of sHsps (and BSA) after 2/4 rounds of desiccation. It also seems necessary to show how sHsps protect CS from inactivation. Do they form complexes with the substrate? If so, what is the size of the complexes? Such analysis is relevant as it needs to be documented whether the protective function of sHsps during desiccation relies on their classical chaperone activity.

To address this concern, we have added SDS-PAGE analysis of soluble and insoluble fractions of citrate synthase in the presence of each supplemental protein after two and four rounds of desiccation (Figure S8). These gels reveal some insolubility of CS after two rounds of desiccation and significantly reduced solubility following four rounds of desiccation. We thank the reviewer for urging us to include this experiment because these data will aid in the interpretation of the activity assay. Consistent with other reports (ex. Goyal *et al.* 2005), we found that A_{340} was highly variable and sometimes reduced after more than two rounds of desiccation. We suspect that the apparent reduction in A_{340} resulted as aggregates coalesced into larger particles that would either completely obstruct or avoid the path of detection. We agree that it would be interesting to look at binding dynamics and more details of the means by which CS is protected. However, it is technically challenging to assess protein-protein interactions in a desiccated state, and further, proteins likely interact nonspecifically during desiccation as they are concentrated in the absence of water. The question remains as to whether or not sHSPs are incorporated into CS aggregates. We did observe sHSPs present in the insoluble fraction along with CS (Figure S8), which suggests that they either bind to or are incorporated into insoluble aggregates. We are hesitant to make specific conclusions about this binding, but further biochemistry that we consider to be beyond the scope of this manuscript may further clarify these mechanisms. Our primary goal in this paper is to provide a proof of principle that sHSPs can limit desiccation-induced aggregation, and the data provided all support this claim.

Figure 5: The authors use BSA as positive control as it was reported to protect proteins against inactivation upon desiccation. The observation that the tested sHsps are similarly active raises the question of specificity as no true negative control (added protein showing no activity) is provided. It is therefore crucial to test for activities of non-related control proteins to exclude that the observed effects simply rely on the presence of high protein concentrations added externally. This will document the specific protective function of sHsps under the given experimental conditions.

Thank you for raising this key point. We have now conducted experiments using two proteins as negative controls. We found that lysozyme did not reduce aggregation of CS (Figure 5e), nor did one of the tardigrade sHSPs that did not impact bacterial desiccation survival (HSP17, Figure 5a). In fact, HSP17 seemingly increased the degree of aggregation of CS (Figure 5a,f). These new data strengthen the argument that HSP21 and HSP24.6 have specific properties that allow them to limit protein aggregation and the reduced aggregation is not just an effect of any general protein additive.

Reviewer #3 (Remarks to the Author):

Hibman et al. reports the capabilities of small HSPs of a tardigrade in enhancing desiccation tolerance when expressed in *E.coli*, and for the protection of enzymatic activities of citrate synthase *in vitro*. This work confirms the previous reports such as Ma et al. 2005 Cryobiology that sHSPs of anhydrobiotic species confer tolerance to desiccation in heterologous systems. (In Ma et al it was mammalian cells and synergistically with trehalose). I especially found Figure S5 to be intriguing, that sHSPs from humans show comparable heterologous protection as tardigrade sHSPs.

We thank the reviewer for these comments. We have more prominently highlighted the work from Ma *et al.* 2005 in our Introduction and more clearly delineate the novelty of our study (Lines 92-93, 97-99). To our knowledge, there have been no studies that have directly shown an effect of sHSPs on protein aggregation *in vitro* (Kim *et al.* 2018, eLife looks at the ability of yeast HSP12 to function as a desiccation-protectant, but despite its name this protein actually an LEA-related protein, not a sHSP – see Mtwisha *et al.* 1998), and there is a general scarcity of studies on tardigrade sHSPs.

My first impression of this work is that while each of the data presented seem interesting, logical connections between the figures seem lacking (the authors do discuss this issue in the Discussions). For example, Fig. 3 shows that HSP21 and HSP24.6 to form small assemblies, but no data is shown regarding the contribution or necessity of this assembly (or even whether this assembly can be seen *in vivo*) in desiccation tolerance. Is this observable in the citrate synthase protection experiment? or in *E.coli*? Would buffer conditions disrupting these oligomerization impair desiccation tolerance?

We appreciate the suggestion to draw links between figures of the paper. We have added to our discussion and improved some transitions. Our goal in providing some data like in Figure 3 is to provide a biochemical characterization of the tardigrade proteins to examine ways in which they may be similar to sHSPs from other organisms. Because virtually nothing is known about tardigrade sHSPs such work provides a novel contribution. Assessing oligomerization has been a common approach – however, we have been unable to find specific conditions to disrupt oligomerization to specifically test the function of oligomer assembly on function during desiccation. It is also possible that post-translational modifications impact oligomerization as has been shown for some other sHSPs (ex Jovcevski *et al.* 2015, Jovcevski *et al.* 2017). Given the lack of information about post-translational modifications of tardigrade sHSPs, we cannot at this stage test this hypothesis. Experiments to test regulation of oligomerization and the functions of monomers vs. oligomers are certainly within our interests moving forward, but we believe them to be beyond the scope of this paper. Our main goal was to determine if tardigrade sHSPs could limit desiccation-induced protein aggregation, which is well-supported by our data.

In Figure 4, only HSP24.6 show increased heat tolerance, and not HSP21, and authors discuss that this may be due to differentiated usage of sHSPs for heat tolerance and for desiccation.

However, HSP24.6 always show better protection from desiccation (in E.coli Fig.2, in vitro Fig.5).

We thank the reviewer for pointing this out. HSP24.6 does show functionality in response to both heat and desiccation. We have qualified our sentence in the Discussion now to say that *some* tardigrade sHSPs may have evolved to function during desiccation (Lines 336-338). It is still the case that no other sHSPs were able to improve bacterial heat shock survival, while multiple proteins could improve desiccation survival. Nonetheless, we have been cautious to not overinterpret these data and provide caveats of this speculation for readers.

In Fig.5, BSA is used as a positive control for protection from protein aggregation under desiccated conditions. Does BSA then improve desiccation tolerance in E.coli? This is not shown in Fig. 2. The key argument of the paper, as the title stands, is the inference of lowered protein aggregation by sHSPs and its effect in desiccation tolerance. However, Fig. 5 only measures A340 readings, which is the NADH consumption by citrate synthase, so it is a measure of enzymatic activity, rather than aggregation. Aggregation can be a reason for the loss in enzymatic activity, but other factors (such as simple denaturation) could also be considered. Lowered protein aggregation in desiccated E.coli is not shown, and it is therefore very difficult for the readers to connect heterologous desiccation protection, protein assembly, and in vitro protection of enzymatic activity, from the presented data.

We agree that the reviewer raises important points. We have now tested if BSA could improve bacterial desiccation survival and found that despite low levels of expression it was still sufficient to improve survival (Figure S6). For the absorbance measurements, we are using light scattering as a measurement for aggregate formation, as routinely done in other papers (for example Goyal *et al.* 2005 and Chakrabortee *et al.* 2012). To ensure that readers understand this, we have clarified this in the figure as well as the text (Figure 5, Lines 522-523). Notably, assessing aggregation by light scattering at 320nm and 500nm provided qualitatively similar results. We then used a commercially available kit to measure the activity of the enzyme as a separate experiment from the aggregation measurements (Figure 5g).

Human sHSP is only shown for the heterologous expression in E.coli. Do they not assemble into oligomeric complexes, and do they not protect citrate synthase from desiccation? If they do, these properties are likely common to sHSP class of proteins, and not to be signified as tardigrade-specific as in the current title and discussions. Moreover, there are different ways these sHSPs can contribute to desiccation tolerance besides protection from protein aggregation, including the protection of membranes (see for example, Sales *et al.* BBA 2000). As described above, I do not see convincing data in the current manuscript to rule out these alternative pathways of protection.

Thank you for these comments and suggestions. We are also interested in understanding correlations between complex assembly and chaperone activity as well as roles for human sHSPs. We further agree that sHSPs likely act in multiple ways to protect cells. However, given the resources required to test these hypotheses we view these questions as beyond the scope of this manuscript. Our primary objective is not to rule out other options, but to demonstrate that one functional property of sHSPs is to limit desiccation-induced aggregation. Our *in vitro* experiments, in which membranes are not present, support this property of sHSPs, and do not rule out other possible properties of sHSPs. To make this clear, we have added to the Discussion to clarify that sHSPs may have diverse means of protection (Lines 393-404).

I suggest to mention more clearly that desiccation tolerance by artemia sHSP in mammalian cells (Ma *et al.* 2005 Cryobiology), increased heat tolerance by them (Liang and MacRae 1999 Dev. Biol), and in vitro protection of cytrate synthase activity by them (Ping *et al.* 1997 Eur. J. Biochem) have already been reported. Current introduction states rather strongly that sHSPs of

animals are not well explored, but very similar experiment to the current manuscript has already been reported for artemia p26. I still see a value in doing similar work for sHSPs from other anhydrobiotic species, namely a tardigrade in this work, but nevertheless the previous efforts should be appropriately noted.

Thank you for these suggestions. We have added to the Introduction to more clearly describe the previous work that has been done and sets the stage for our analyses (Lines 90-93). We believe it is still appropriate to describe roles for sHSPs in animal desiccation tolerance as less well-explored than roles in heat tolerance given these limited studies.

Reviewers' comments:

Reviewer #1 (Remarks to the Author):

In their revised version the authors have partially addressed my previous concerns. Determining the *in vivo* solubilities of tardigrade and human sHsps is appreciated as this also contributes to the understanding why some sHsps do not protect *E. coli* cells from desiccation. The addition of negative controls in the *in vitro* desiccation experiments and the determination of Citrate Synthase (CS) solubilities after several rounds of desiccation is also acknowledged, however, the particular assay and the drawn conclusions remain problematic for the following reasons:

Figure S8, which reports on CS solubilities after two and four rounds of desiccation, does not reveal differences between absence and presence of sHsps. After 4 rounds of desiccation CS is almost quantitatively found in the pellet fraction and the addition of sHsps but also the positive control BSA does not change that. This observation is in conflict with CS aggregation data based on absorbance (A₃₄₀) measurements (Fig. 5). This discrepancy is not discussed by the authors, in fact Figure S8 and respective data are not mentioned in the text. Determining protein aggregation based on A₃₄₀ measurements therefore seems problematic. It is also noted that Hsp17 and Hsp21, which either increases CS aggregation (Hsp17, based on A₃₄₀) or does not significantly change substrate aggregation (Hsp21), both enhance CS activity. CS protection therefore cannot be correlated with changes in A₃₄₀-based aggregation measurements. The drawn conclusions drawn from this assay are therefore problematic. While the CS activity measurements (Fig. 5g) demonstrate a protective function of sHsps during desiccation, the specificity of this activity (Hsp17 also shows activity) and its molecular basis remains ill defined. In view of the points raised above the manuscript is in need of further revision/clarification.

Further point:

Lane 223 (Figure S4): There is no difference between Hsp21 and the negative control BSA. The assay therefore does not document the ability of Hsp21 to suppress CS aggregation at high temperatures.

Reviewer #3 (Remarks to the Author):

The additional experiment for increased desiccation tolerance of BSA-expressing *E. coli* is an interesting data. So BSA and human sHSP has similar level of desiccation tolerance enhancement as tardigrade sHSPs in *E. coli*, and BSA has similar or seemingly higher protective capabilities during desiccation cycles of Citrate Synthase as/than tardigrade sHSPs. I think the series of data is showing that the use of *E. coli* is not optimal in probing desiccation tolerance mechanisms of animals, and that tardigrade sHSPs are in deed sHSPs, not too special compared to the human counterparts (at least direct comparison showing superiority of tardigrade sHSPs is lacking). The authors state in their rebuttal letter that there have been no reports on the study of tardigrade sHSPs, which is true, and I agree that there is a value in reporting that tardigrade sHSPs, likewise sHSPs found in other organisms, oligomerize into large complexes, and protects *E. coli* and enzymes as much as human sHSPs or BSA does, and that they are not much different from sHSPs from other animals, but the current abstract reads as if tardigrade sHSPs are special, which seems misleading.

Reviewers' comments:

Reviewer #1 (Remarks to the Author):

In their revised version the authors have partially addressed my previous concerns. Determining the in vivo solubilities of tardigrade and human sHsps is appreciated as this also contributes to the understanding why some sHsps do not protect *E. coli* cells from desiccation. The addition of negative controls in the in vitro desiccation experiments and the determination of Citrate Synthase (CS) solubilities after several rounds of desiccation is also acknowledged, however, the particular assay and the drawn conclusions remain problematic for the following reasons:

Figure S8, which reports on CS solubilities after two and four rounds of desiccation, does not reveal differences between absence and presence of sHsps. After 4 rounds of desiccation CS is almost quantitatively found in the pellet fraction and the addition of sHsps but also the positive control BSA does not change that. This observation is in conflict with CS aggregation data based on absorbance (A₃₄₀) measurements (Fig. 5). This discrepancy is not discussed by the authors, in fact Figure S8 and respective data are not mentioned in the text. Determining protein aggregation based on A₃₄₀ measurements therefore seems problematic. It is also noted that Hsp17 and Hsp21, which either increases CS aggregation (Hsp17, based on A₃₄₀) or does not significantly change substrate aggregation (Hsp21), both enhance CS activity. CS protection therefore cannot be correlated with changes in A₃₄₀-based aggregation measurements. The drawn conclusions drawn from this assay are therefore problematic. While the CS activity measurements (Fig. 5g) demonstrate a protective function of sHsps during desiccation, the specificity of this activity (Hsp17 also shows activity) and its molecular basis remains ill defined. In view of the points raised above the manuscript is in need of further revision/clarification.

The reviewer is correct that any differences in solubility of CS are modest. Because the samples were run across multiple gels, quantitative comparisons are not possible. However, the limited differences in solubility do not invalidate the results from A₃₄₀ readings. There need not be a direct correlation between aggregation and solubility for a couple reasons: 1) some aggregates may scatter light while they retain solubility which could give positive readings by A₃₄₀ while not precipitating to the insoluble fraction, 2) aggregates can be of variable size and density which can impact light scattering measurements, while at the same time the entire population of protein may have a similar level of solubility. We suspect that this second option may give rise to the result the reviewer notes – namely, that aggregation is seemingly reduced in some experimental conditions like addition of BSA or HSP24.6 despite a lack of clear differences in solubility. In other words, addition of BSA or HSP24.6 may lead to smaller but more numerous aggregates relative to fewer large aggregates in other conditions. Notably, prior papers that established standard assays for protein aggregation have repeatedly used light scattering as a readout for aggregation (for example Jakob *et al.* 1993, Goyal *et al.* 2005, Chakrabortee *et al.* 2007). While recent studies have shown that aggregation alone can be either harmful or protective in a context-dependent manner, assessing enzyme activity gives a direct functional readout of protein protection. Thus, by measuring enzyme activity as a direct method to assess protection we are confident in the conclusion that HSP17, HSP21, HSP24.6, and BSA are protective by this definition. We regret that we failed to mention Figure S8 in the text and have now added this reference and revised our presentation of the results to clarify the interpretation and conclusions (Lines 297-301).

The reviewer further highlights the complexity of HSP17 which increased levels of aggregation, but was still slightly protective at the highest concentration tested. The above point that aggregation alone may be helpful or harmful may well be relevant here. We speculate that

HSP17 may act as a sequestrase, an evolutionarily conserved property of some sHSPs (Lines 309-313). These sHSPs can generate inclusion bodies with substrate that are actually protective. Because studying HSP17 was not the goal of this work and due to the resources that would be required to unequivocally demonstrate sequestrase function, we prefer to be transparent in reporting our results and to clarify in the text the limits of the conclusions we make. We have now more explicitly stated in the results that measuring enzyme activity is the most direct readout of protective ability (Lines 302-304). These revisions clarify the conclusions we draw.

Further point:

Lane 223 (Figure S4): There is no difference between Hsp21 and the negative control BSA. The assay therefore does not document the ability of Hsp21 to suppress CS aggregation at high temperatures.

We have added clarification to the results to indicate that HSP21 is less effective than HSP24.6 and in fact no different than BSA in its ability to chaperone at a higher concentration (Lines 223-225).

Reviewer #3 (Remarks to the Author):

The additional experiment for increased desiccation tolerance of BSA-expressing *E. coli* is an interesting data. So BSA and human sHSP has similar level of desiccation tolerance enhancement as tardigrade sHSPs in *E. coli*, and BSA has similar or seemingly higher protective capabilities during desiccation cycles of Citrate Synthase as/than tardigrade sHSPs. I think the series of data is showing that the use of *E. coli* is not optimal in probing desiccation tolerance mechanisms of animals, and that tardigrade sHSPs are in deed sHSPs, not too special compared to the human counterparts (at least direct comparison showing superiority of tardigrade sHSPs is lacking). The authors state in their rebuttal letter that there have been no reports on the study of tardigrade sHSPs, which is true, and I agree that there is a value in reporting that tardigrade sHSPs, likewise sHSPs found in other organisms, oligomerize into large complexes, and protects *E. coli* and enzymes as much as human sHSPs or BSA does, and that they are not much different from sHSPs from other animals, but the current abstract reads as if tardigrade sHSPs are special, which seems misleading.

We appreciate the reviewer's recognition that the new data from BSA-expressing *E. coli* are consistent with its protective role. While heterologous expression experiments have limits in terms of the relevance to *in vivo* biology, we are clear about these limits throughout the text and interpret results from these experiments that test sufficiency to conclude that desiccation protection is a property that sHSPs *can* have. This language is reflected in the title to suggest that sHSPs can function during desiccation. The proof of principle that sHSPs can act in this manner is still a novel discovery and sets the stage for future work to test the extent to which they are utilized as such protectants *in vivo*, and in combination with other endogenous mechanisms of protection.

We are grateful for the opportunity to clarify that desicco-protection may be a conserved property of sHSPs and not unique to tardigrades. We have edited the abstract to state this more clearly ("Multiple tardigrade and human sHSPs could improve desiccation tolerance of *E. coli*, suggesting that the capacity to contribute to desicco-protection is a conserved property of some sHSPs.").

REVIEWERS' COMMENTS:

Reviewer #1 (Remarks to the Author):

The authors have sufficiently addressed my former concerns and comments in their newly revised version. The added clarifications on substrate turbidity, solubility and activity measurements are appreciated and improve data interpretation. The described protective function of sHsps in dessication-induced protein aggregation is exciting and will stimulate further research and justifies publication in Communications Biology.